# Adaptive Deep Brain Stimulation (aDBS) for Tourette Syndrome

**DOI:** 10.3390/brainsci8010004

**Published:** 2017-12-23

**Authors:** Sara Marceglia, Manuela Rosa, Domenico Servello, Mauro Porta, Sergio Barbieri, Elena Moro, Alberto Priori

**Affiliations:** 1Clinical Center for Neurostimulation, Neurotechnology and Movement Disorders, Fondazione Istituto Ricovero e Cura a Carattere Scientifico (IRCCS) Ca’ Granda, Ospedale Maggiore Policlinico, Milan 20122, Italy; manuela.rosa@policlinico.mi.it (M.R.); sergio.barbieri@policlinico.mi.it (S.B.); 2Dipartimento di Ingegneria e Architettura, Università degli Studi di Trieste, Trieste 34127, Italy; 3Functional Neurosurgery Unit, Galeazzi Hospital and Tourette Center, Milan 20161, Italy; servello@libero.it (D.S.); mauroportamilano@gmail.com (M.P.); 4Division of Neurology, Centre Hospitalier Universitaire de Grenoble, CS 10217, 38043 Grenoble, France; emoro@chu-grenoble.fr; 5“Aldo Ravelli” Center for Neurotechnology and Experimental Brain Therapeutics, University of Milan, Milan 20142 , Italy; alberto.priori@unimi.it; 6Department of Health Sciences, University of Milan & ASST Santi Paolo e Carlo, Milan 20142, Italy

**Keywords:** Tourette syndrome, adaptive deep brain stimulation, local field potentials, electrophysiology, thalamus, nucleus accumbens, closed-loop

## Abstract

Deep brain stimulation (DBS) has emerged as a novel therapy for the treatment of several movement and neuropsychiatric disorders, and may also be suitable for the treatment of Tourette syndrome (TS). The main DBS targets used to date in patients with TS are located within the basal ganglia-thalamo-cortical circuit involved in the pathophysiology of this syndrome. They include the ventralis oralis/centromedian-parafascicular (Vo/CM-Pf) nucleus of the thalamus and the nucleus accumbens. Current DBS treatments deliver continuous electrical stimulation and are not designed to adapt to the patient’s symptoms, thereby contributing to unwanted side effects. Moreover, continuous DBS can lead to rapid battery depletion, which necessitates frequent battery replacement surgeries. Adaptive deep brain stimulation (aDBS), which is controlled based on neurophysiological biomarkers, is considered one of the most promising approaches to optimize clinical benefits and to limit the side effects of DBS. aDBS consists of a closed-loop system designed to measure and analyse a control variable reflecting the patient’s clinical condition and to modify on-line stimulation settings to improve treatment efficacy. Local field potentials (LFPs), which are sums of pre- and post-synaptic activity arising from large neuronal populations, directly recorded from electrodes implanted for DBS can theoretically represent a reliable correlate of clinical status in patients with TS. The well-established LFP-clinical correlations in patients with Parkinson’s disease reported in the last few years provide the rationale for developing and implementing new aDBS devices whose efficacies are under evaluation in humans. Only a few studies have investigated LFP activity recorded from DBS target structures and the relationship of this activity to clinical symptoms in TS. Here, we review the available literature supporting the feasibility of an LFP-based aDBS approach in patients with TS. In addition, to increase such knowledge, we report explorative findings regarding LFP data recently acquired and analysed in patients with TS after DBS electrode implantation at rest, during voluntary and involuntary movements (tics), and during ongoing DBS. Data available up to now suggest that patients with TS have oscillatory patterns specifically associated with the part of the brain they are recorded from, and thereby with clinical manifestations. The Vo/CM-Pf nucleus of the thalamus is involved in movement execution and the pathophysiology of TS. Moreover, the oscillatory patterns in TS are specifically modulated by DBS treatment, as reflected by improvements in TS symptoms. These findings suggest that LFPs recorded from DBS targets may be used to control new aDBS devices capable of adaptive stimulation responsive to the symptoms of TS.

## 1. Introduction

Tourette syndrome (TS) is a complex condition characterized by tics and is often accompanied by psychiatric comorbidities. The treatment of TS is thus still challenging, especially in a subset of drug-refractory and severely affected patients [1]. In the last 15 years, deep brain stimulation (DBS) has emerged as a promising therapeutic intervention for TS, with more than 100 patients with TS implanted worldwide [2].

DBS consists of the placement of deep brain electrodes in a target structure. These electrodes are connected to an implanted pulse generator that delivers high-frequency (>100 Hz) electrical stimulation. While the optimal target has already been determined in other pathologies, such as dystonia, essential tremor, and Parkinson’s disease [3], several different targets have been reported for TS [2,4]. The stimulation of these targets has led to both success and failure. The medial part of the thalamus [2] (centro-median nucleus (CM), ventralis oralis nucleus (Vo), and parafascicular nucleus (Pf)), the globus pallidus internus [5] (GPi, anterior and posteroventrolateral parts), the internal capsule, the nucleus accumbens (NA), and the fields of Forel (H1) [6] have all been explored as DBS targets in TS. Findings from the different studies reported in the literature suggest that, apart from network considerations, the choice of target should be driven by the potential to reduce the different comorbidities associated with TS in addition to the aim of reducing tics [7]. In addition to differences in target selection, differences in the age of inclusion contribute to the heterogeneity of the available results [8]. In the long term (7–9 years), thalamic DBS led to a 50% reduction in tics, suggesting an overall efficacy in patients with severe refractory TS [9]. This type of stimulation is thus thought to have the potential to improve social functioning [10]. However, DBS is often associated with post-surgery complications [2] that lead to imbalances between side effects and therapeutic effects, and often leads to the switching off of the stimulator or to new surgeries for the placement of additional electrodes in different brain areas [11].

TS is a movement and neuropsychiatric disorder characterized by fluctuation, while DBS is delivered continuously, in a constant manner, without any adaptation of the stimulation paradigm to the tic dynamic. This is the same line of reasoning applied to DBS for the treatment of Parkinson’s disease. This has led to the proposal of novel closed-loop, adaptive DBS (aDBS) approaches to change stimulation parameters based on the patient’s clinical state [12,13,14]. In Parkinson’s disease, the aDBS approach tested in humans is based on a strategy of adapting stimulation parameters (amplitude) based on changes in the patterns of local neuronal activity (i.e., local field potentials, LFPs) recorded through the implanted DBS electrode while DBS is turned on. The large body of literature showing that LFP patterns are representative of the clinical state in Parkinson’s disease has guided the technological development of aDBS.

The only alternative to classical continuous DBS for the treatment of TS proposed up to now is “intermittent” stimulation [15,16]. In the proposed “scheduled” paradigm, DBS is delivered following an a priori determined schedule that decreases the time during which DBS is switched on. In fact, an average of 2.3 h/day of stimulation has been shown to lead to significant improvement in the Yale Global Tic Severity Scale after 2 years [15].

Considering that LFPs have also been recorded and analysed from DBS electrodes in patients with TS, and that the LFP patterns have specific characteristics during tics, it can be hypothesized that LFP-based aDBS approaches may also be used for the treatment of refractory TS [17,18,19,20,21]. Here we provide a review of the available literature supporting the feasibility of an LFP-based aDBS approach for the treatment of TS. To further support this hypothesis, we also report explorative findings regarding LFPs recently recorded in patients with TS after DBS electrode implantation. After a short explanation of the concept of adaptive DBS guided by LFPs, we discuss the specific LFP signature in TS at rest, during voluntary and involuntary movements (tics), and during ongoing DBS in separate sections of the manuscript. The discussion section summarizes the results reported in the literature and those of the additional experiments reported here, and considers the feasibility of aDBS in patients with TS.

## 2. The LFP-Based aDBS Concept

Adaptive DBS is based on a model whereby the implanted aDBS device measures and analyses a variable (called the “control variable”) that correlates with the patient’s clinical state. The device then changes the DBS parameters according to these data to provide optimal moment-by-moment stimulation to control the patient’s symptoms. The new stimulation paradigm, which would change the patient’s state, would in turn induce a change in the control variable, which is measured and analysed again, thus closing the loop. In aDBS, the choice of the control variable should take into account some important technological, practical, and clinical requirements. More specifically, the control variable should be measured without the need for additional implants and must reflect the patient’s state. In addition, the processing and computational costs of the analysis should be kept low to ensure low power consumption. Finally, the implantation of an aDBS system should not change normal clinical practice, and it should not impact the patient’s acceptance [22].

Compound pre-synaptic and post-synaptic activities of the local neuronal populations recorded using DBS electrodes, named LFPs, have been chosen as the most promising control variable for aDBS, at least in Parkinson’s disease [22]. LFPs recorded from one (monopolar recording) or two (bipolar recording) contacts on the implanted DBS electrode do not require additional implants or any specific changes to neurosurgical procedures. Moreover, as the aDBS technology is included in the same implantable device (the only change is inside the neurostimulator), the level of acceptance of LFP-based aDBS is the same as that of DBS. The bulk of evidence collected in almost 20 years of research on LFPs indicates that they are characterised by oscillations through which they encode information within the cortico-basal ganglia-thalamo-cortical loop. LFP oscillations occur over a wide frequency spectrum and include the very low-frequency band (range: 2–7 Hz), alpha/low beta band (range: 8–20 Hz), high beta band (range: 20–35 Hz), gamma band (range: 60–80 Hz), and very high-frequency band (range: 250–350 Hz). Each oscillation band is specifically modulated by changes in the patient’s clinical state and movement execution, and by cognitive and behavioural stimuli [22,23,24,25].

The large majority of LFP studies have been conducted in the short time window between stereotactic neurosurgery for electrode implantation and the implantation of the subcutaneous pulse generator, known as the “acute” phase. However, since it is a transient and continuously evolving condition, this acute phase cannot be representative of the “chronic” condition during which DBS is delivered. Therefore, other experiments have been conducted in patients in need of battery replacement surgery after up to 7 years of DBS [26]. These experiments have been crucial in establishing that LFPs are consistently recordable over time, thus confirming the feasibility of LFP-based aDBS.

LFPs have been recorded not only in patients with Parkinson’s disease, but also in those with other movement and neuropsychiatric disorders treated with DBS, including Tourette syndrome. Different LFP patterns characterize different pathologies and conditions, suggesting that an LFP-based adaptive strategy is feasible not only in Parkinson’s disease, but also in other conditions.

### 2.1. Experiment 1: LFPs in Tourette Syndrome at Rest

LFP recordings obtained at rest from the thalamus in patients with TS during tic-free epochs are characterized by activity predominantly at low-frequencies (2–7 Hz) and in the alpha band (8–13 Hz), but not in the beta band (20–35 Hz) [20]. Increased low-frequency intra-thalamic coherence has been reported to not be time-locked with motor tics [18].

This LFP pattern is consistent with the hypothesis that increased bursting of thalamic cells in the range of the low-frequency band generates so-called thalamo-cortical dysrhythmia, which in turn results in increased coherence between low-frequency and high-frequency rhythms [27]. The low-frequency pattern observed in TS also resembles the increase within the low-frequency band across the basal ganglia-thalamo-cortical loop, which characterizes hyperkinetic disorders, such as dystonia or parkinsonian dyskinesias [28,29].

Speculation that unbalanced low-frequency activity contributes to the hyperkinetic signature of TS is further supported by LFP recordings in the GPi in patients with Tourette syndrome. These recordings have revealed a consistent increase in low-frequency activity, together with low-frequency/high-frequency synchronization (200–400 Hz) [17]. We therefore conducted additional experiments to further confirm the reported LFP patterns of DBS targets in patients with TS.

#### 2.1.1. Methods and Participants

To further confirm the above-mentioned results, we studied 7 patients with severe TS (5 men and 2 women, age range: 24–47 years, Table 1) refractory to standard drug treatment, satisfying the DSM-IV-TR (American Psychiatric Association, APA 2000) and World Health Organization criteria for TS (WHO 1992). The patients underwent DBS surgery at the Functional Neurosurgery Unit at IRCCS Galeazzi hospital. The patients were bilaterally implanted with macroelectrodes for DBS (model 3389, Medtronic; Minneapolis, MN, USA) either in the ventralis oralis/centromedian-parafascicular (Vo/CM-Pf) nucleus of the thalamus or the NA (see Table 1 for clinical and DBS details). The surgical procedure is fully described elsewhere [20,30]. The study was approved by the local Ethical Committee of the IRCCS Galeazzi hospital and conformed to the declaration of Helsinki. All patients provided written informed consent. We recorded LFPs during the “acute” phase 4–5 days after surgery for electrode placement. LFP recordings at rest (patients seated on an armchair) were captured bipolarly from the available contact pairs (0–1, 1–2, and 2–3), first from the right electrode and then from the left electrode. Signals were acquired using a Galileo BE Light electroencephalography (EEG) amplification system (EBNeuro Spa; Florence, Italy) with a 2–500 Hz band pass filter, a 1024 Hz sampling frequency, and 12-bit quantization over a 5 V range. Data were analysed off-line using Matlab software (version 7.10, The MathWorks; Natick, MA, USA). After preliminary 125 Hz resampling, oscillatory activity was quantified in the frequency domain by analysing the power spectral density (PSD) using Welch’s averaged modified periodogram with a resolution of 1 Hz [31].

#### 2.1.2. Results

LFPs recorded from the Vo/CM-Pf nuclei (*n* = 8) revealed two main oscillatory activity patterns: one in the low-frequency band (2–7 Hz) and one in the alpha band (8–13 Hz). No spectral peak in the beta band was detected in any nucleus. The low-frequency band intensity in the LFP recordings obtained from contacts 0–1 was higher than that obtained from the other contacts (Figure 1A). It was possible to isolate only one main oscillation pattern in the LFPs recorded from the NA (*n* = 4). This activity in the low-frequency band was higher in recordings from the most caudal contacts (0–1) than in those from the other contacts (Figure 1B).

### 2.2. Experiment 2: LFPs in Tourette Syndrome during Voluntary Movements

LFP recordings from the thalamus [18] and the GPi [17] during voluntary movement have been reported in a limited number of patients. In contrast to tics, voluntary movements are characterized by cortical involvement represented by a pre-motor potential arising before movement execution [18], and by increased alpha activity in the GPi [17].

#### 2.2.1. Methods and Participants

To build on the previously reported observations, we recorded bilateral LFPs during both self-paced and externally-cued movements of the upper limbs in the same 7 patients with TS described in the previous section (Table 1). The patients were instructed to lift their upper limb every 10 s as quickly as possible 15 times (right limb first, LFPs recorded bilaterally). To monitor the movements, electromyography (EMG) signals were recorded from the anterior deltoid muscle using a pair of Ag/AgCl electrodes. After a rest period to avoid fatigue, the patients were instructed to lift their upper limb after an external cue (right limb first). To determine the time course of single oscillatory bands during the movements, we used the squared module of the Hilbert transform applied to LFPs band-passed in each band of interest (low-frequency (2–7 Hz) alpha (8–13 Hz), and high beta (20–35 Hz)), as previously described [32].

#### 2.2.2. Results

We excluded some LFP traces from the analysis due to movement artefacts that reduced the quality of the signals. Based on the results reported for the GPi, the most consistent power modulations during voluntary movements occur in the low-frequency and alpha bands. Low-frequency power increased during the movement phase for both externally-cued and self-paced movements. The low-frequency power increase was more prominent for externally-cued movements than for self-paced movements, and began before movement onset (pre-movement phase) only for externally-cued movements. Similarly, the alpha power modulation had an increase in the pre-movement phase only for the externally-cued movements (Figure 2A,B). Table 2 details the average percentage changes and the timing of the observed modulations. 

### 2.3. Experiment 3: LFPs during Involuntary Movements

Studying LFP activity during tics is one of the most important steps in establishing whether aDBS is feasible in patients with Tourette syndrome. However, the immediate post-surgical lesion effect likely decreases the occurrence of tics in patients with TS undergoing DBS. This limits the possibility of obtaining significant data using acute LFP recordings. In addition, the presence of the sensory phenomenon of feeling an “urge to move” (premonitory urge), which is relieved by tic expression, has been documented as a distinct feature of TS [33]. This suggests the need to study the neurophysiological correlates of such phenomena to better understand the phenomenology of TS tics. However, neuroimaging studies have reported that premonitory urge is mainly associated with activation of the neocortical and paralimbic areas, and some activation in the putamen, caudate, and claustrum [34]. There are no LFP studies specifically reporting observations regarding premonitory urge.

Increased gamma (35–200 Hz) and high-frequency oscillations (200–400 Hz) in the GPi have been reported during tics. These changes are associated with synchronization of the high-frequency activity and the beta band [17], which is consistent with cortical beta desynchronization before tics [19]. In the thalamus, low-frequency (below 10 Hz) and high-frequency (30–100 Hz) activity is higher during tics [19]. This is associated with increased thalamocortical coherence observed immediately before tic onset [18].

Shute and colleagues [19] have used the Medtronic Activa PC implanted pulse generator (IPG) with stimulation disabled during data collection. Use of this implanted device allowed the authors to overcome the post-surgical limitations of acute recordings. Interestingly, the authors reported that, using the increased LF activity, tics had a highly detectable thalamocortical signature with an average sensitivity of 88.6% and an average precision of 96.3% over a 6-month period. However, this study used cortical electrodes to increase the precision of tic detection, which required the inclusion of an adjunctive device to be implanted. This represents a limitation of the aDBS approach.

#### 2.3.1. Results

In our sample of 7 patients with TS undergoing DBS, only one patient with the implant in the Vo/CM-Pf showed tics during recording in the acute phase (patient no. 3 in Table 1). The patient had severe motor tics in both the upper and lower limbs. In this patient, LFP recordings revealed two main oscillatory activity patterns that changed during tics: one in the low-frequency band and the other in the alpha band (Figure 3A). Low-frequency power increased from baseline during both the movement phase and the recovery phase for both the upper and lower limb tics. Alpha power also increased from baseline during the movement and recovery phases for the lower limb tics, and only during the movement phase for the upper limb tics (Figure 3B,C). The alpha band also had desynchronization in the pre-movement phase, particularly during the upper-limb tics. This desynchronization, even though scarcer and anticipated in time, was also observable during the lower-limb tics. Table 2 details the average percentage changes and timing of the observed modulations.

### 2.4. Experiment 4: Chronic LFP Recordings during DBS

There are no reports in the literature of ipsilateral LFP recordings from the same electrode during DBS in patients with TS, likely due to certain technical challenges needing to be addressed in order to guarantee accurate recordings [18,35]. We have previously developed a device able to record LFPs from the electrode delivering DBS. We have used this device to obtain LFP recordings from patients with Parkinson’s disease during stimulation from the same electrode. This device is called *FilterDBS* (Newronika srl, Milan, Italy) [35].

#### 2.4.1. Methods and Participants

Using *FilterDBS*, we recorded LFPs at rest (baseline, 3 min), during ongoing DBS (DBS ON, 6 min), and after the DBS was turned off (post-DBS, 3 min). We performed the recordings in 8 patients with severe TS (6 men and 2 women, age range: 24–48 years) treated with DBS for 1–7 years (Table 3) who were in need of battery replacement. Using the above strategy, we avoided the lesional effects typical of acute recordings and demonstrated that LFPs are consistently recordable years after DBS. The study was approved by the local Ethical Committee of the IRCCS Galeazzi hospital and conformed to the declaration of Helsinki. All patients provided written informed consent.

LFP recordings took place in the surgery room while the lead extensions were exposed before connection to the new implanted stimulator. Therefore, the experimental setting limited our recordings to those obtained at rest (no movement-related or tic-related LFPs were recorded) with and without DBS turned on.

#### 2.4.2. Results

We observed that DBS modulates LFP oscillatory activity by specifically increasing low-frequency power from baseline. The other power frequencies remained unchanged. After DBS was turned off, LF power decreased and returned to the baseline LF power value (Figure 4A,C). In one patient (patient no. 5, Table 3), we also observed a decrease in the alpha band when DBS was turned on. The alpha band value returned to baseline after DBS was turned off (Figure 4B,D).

We hypothesize that DBS saturates low-frequency oscillations, which in turn are no longer available for the thalamo-cortical communication characterizing tics. This effect, which occurs ipsilateral to the site of stimulation, may work in synergy with the suppression of the alpha rhythm observed in one patient, which has consistently been observed on the side contralateral to the stimulation in other groups [18].

## 3. Discussion

The results reported in the literature, together with the original results reported here, suggest that LFP-based aDBS may be feasible in patients with TS. These results, however, have been obtained mainly in the thalamus, which is one of the first targets explored for DBS in TS. This limits the possibility of designing LFP-based aDBS algorithms dedicated to other targets. This is an area of research requiring further experiments.

In support of the feasibility of LFP-based aDBS, we know that thalamic LFPs show consistent modulations in patients with TS, particularly in the LF (2–7 Hz) and alpha (8–13 Hz) bands during tics, during voluntary movements, and during DBS. In addition, thalamic low-frequency modulation has been shown to be effective as a readout for recognizing tics [19]. Involuntary movements are characterized by a decrease in alpha band activity (−20%) for 250 ms, which is followed by a large increase (+150%) in both LF and alpha band activity for at least 1 s (Table 2). Conversely, voluntary movements are characterised by increased LF band activity (+40%) and a small increase (maximum of 20%) in alpha band activity for 250 ms, followed by an increase (+60%) in both LF and alpha band activity for at least 500 ms (Table 2). We provided evidence that LFPs can still be recorded after years of stimulation and when DBS is turned on without the need for additional implants. This supports the feasibility of chronic LFP-based aDBS in patients with TS. Finally, in contrast to the classical feedback approach used in Parkinson’s disease [13,14,22], the findings obtained in patients with TS suggest that a feed-forward strategy may be preferable [16].

If the observations made in our limited sample are confirmed in larger studies, we would speculate that, in patients implanted with thalamic DBS devices, monitoring of thalamic alpha and LF power changes in windows of 250 ms should be sufficient to detect significant changes in the oscillatory pattern. We would then be able to use these changes to identify tics and to change specific DBS parameters. As an example of the potential applicability of an LFP-based aDBS approach, Figure 5 shows an adaptive feed-forward strategy to change DBS parameters when tic onset is potentially detected based on the available LFP characteristics during involuntary movements. Based on this hypothesis, DBS may be delivered continuously with the optimized parameters set for each patient (usually 130 Hz with a 2–5 V amplitude range and a 60–120 μs pulse-width range). LFPs are recorded during DBS. The adaptive system may extract the LF and alpha bands by applying a band-pass filter (one for LF at 2–7 Hz and one for alpha at 8–13 Hz) followed by a 100 ms moving average filter. This would allow us to observe LF and alpha changes over 100 ms with a 100 ms delay. These changes can thus be detected in the time scale required for the detection of fast movement-related dynamics.

When a decrease in the alpha power of at least 20% lasting for at least 250 ms is detected, the system verifies whether it is followed by a 150% increase in both the alpha and LF signals. If this is the case, the DBS parameters are changed to decrease the LF signal and suppress the alpha changes. We hypothesize that a decrease in the pulse width to 10–30 μs and a decrease in the frequency around the high beta band (30–35 Hz) would help stop involuntary movements and decrease the LF and alpha band activities. When the LF band returns to a maximum of +50%, the DBS can be switched to the classical optimized parameters.

However, the scheme presented above is only a theoretical speculation based on the literature and the current findings regarding tic detection and DBS effects on LFPs recorded in the thalamus in patients with Tourette syndrome. The above approach has several important limitations. First, the neurophysiological observations underlying the adaptive model were obtained in a limited number of patients and should be confirmed in larger studies before they are considered conclusive. In addition, tic phenomenology should take into account the presence of the premonitory urge [33], which would further enhance the ability of the feed-forward strategy to suppress tics. In fact, if the premonitory urge is recognised in neurophysiological recordings, it may be used as a trigger for specific DBS patterns aimed to suppress the urge, which would otherwise be satisfied by tic expression. This would, in effect, prevent tic onset. This hypothesis is at present purely speculative and would require targeted experiments to specifically study the LFP signature (if any) of the premonitory urge and the ability of DBS to satisfy the urge.

Finally, this first exploratory model for aDBS is based on data obtained from the thalamus and grounding on previous technology developed for aDBS in Parkinson’s Disease [36]. As such, it does not take into account psychiatric comorbidities that are observed in a large number of patients with TS [37], which have recently led to the characterization of obsessive-compulsive tic disorder, which is a reflection of obsessive-compulsive disorder and TS comorbidity [38]. The patients analysed here were, for the most part, implanted in the Vo/CM-Pf thalamic complex. However, as for Parkinson’s disease, LFP studies for TS have first focused on motor symptoms and then on cognitive dysfunctions [22]. Therefore, to propose an advanced aDBS model, new experiments specifically designed to evaluate LFPs in domains such as attention, memory, executive function, language, motor, and visuomotor functions should be conducted, especially in patients implanted in target areas, such as the anterior GPi [38].

## 4. Conclusions

In conclusion, the present review, which was enriched by original results, provides some indications supporting the hypothesis that LFP-based aDBS, at least in the thalamus, is feasible in TS. Further studies in patients are required to better understand the LFP signatures of psychiatric comorbidities and non-tic disease manifestations. These studies should include other DBS targets that are now considered very promising for the treatment of TS, such as the anterior GPi.

## Figures and Tables

**Figure 1 brainsci-08-00004-f001:**
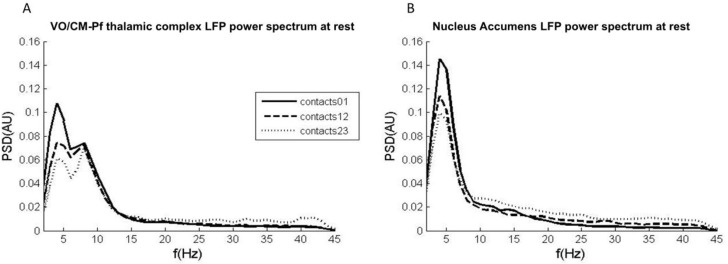
LFP power spectrum at rest obtained from the Vo/CM-Pf and NA nuclei. (**A**) Average power spectrum across all Vo/CM-Pf nuclei (*n* = 8) obtained from LFPs captured from contacts 0–1 (black solid line), 12 (dashed black line), and 23 (dashed grey line). The *x*-axis represents frequency (Hz) and the *y*-axis represents the normalized power spectral density (PSD, arbitrary units). (**B**) Average power spectrum across all NA nuclei (*n* = 4) obtained from LFPs captured from contacts 0–1, 1–2, and 2–3. The plot is organised as it is in panel A.

**Figure 2 brainsci-08-00004-f002:**
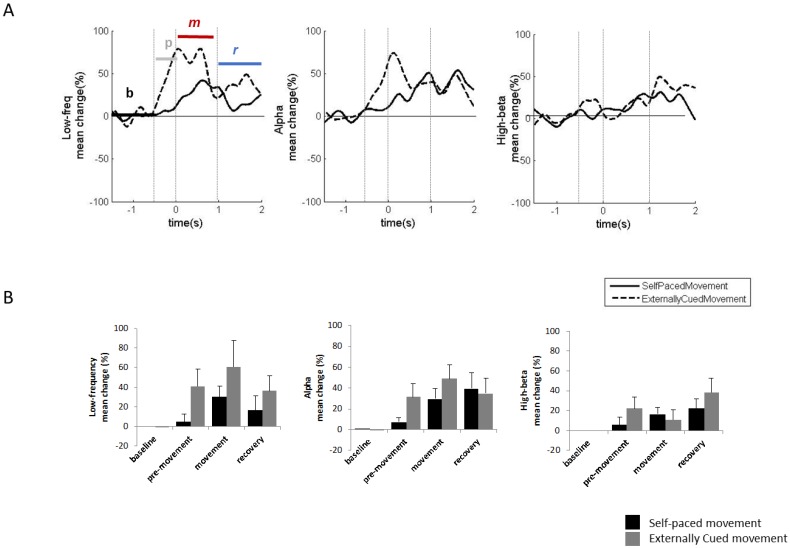
Vo/CM-Pf LFP power modulation during voluntary movements. (**A**) Grand average (*n* = 7) of the low-frequency (**left plot**), alpha (**central plot**), and high beta (**right plot**) power modulations during self-paced voluntary movements (solid black line) and externally-cued movements (dashed black line). Power modulations are expressed as percentage changes from the baseline phase and were estimated starting from 1.5 s before the movement onset until 2 s after the movement onset. The four movement-related phases were baseline (b), pre-movement (p), movement (m), and recovery (r). *x*-axis: time (s); *y*-axis: Hilbert-power modulations (Percentage). (**B**) The histograms represent the mean power modulation for low-frequency (**left plot**), alpha (**central plot**), and high beta (**right plot**) bands in the four movement phases (baseline, pre-movement, movement, and recovery) for self-paced movements (black bars) and externally-cued movements (grey bars). Error bars represent the standard error.

**Figure 3 brainsci-08-00004-f003:**
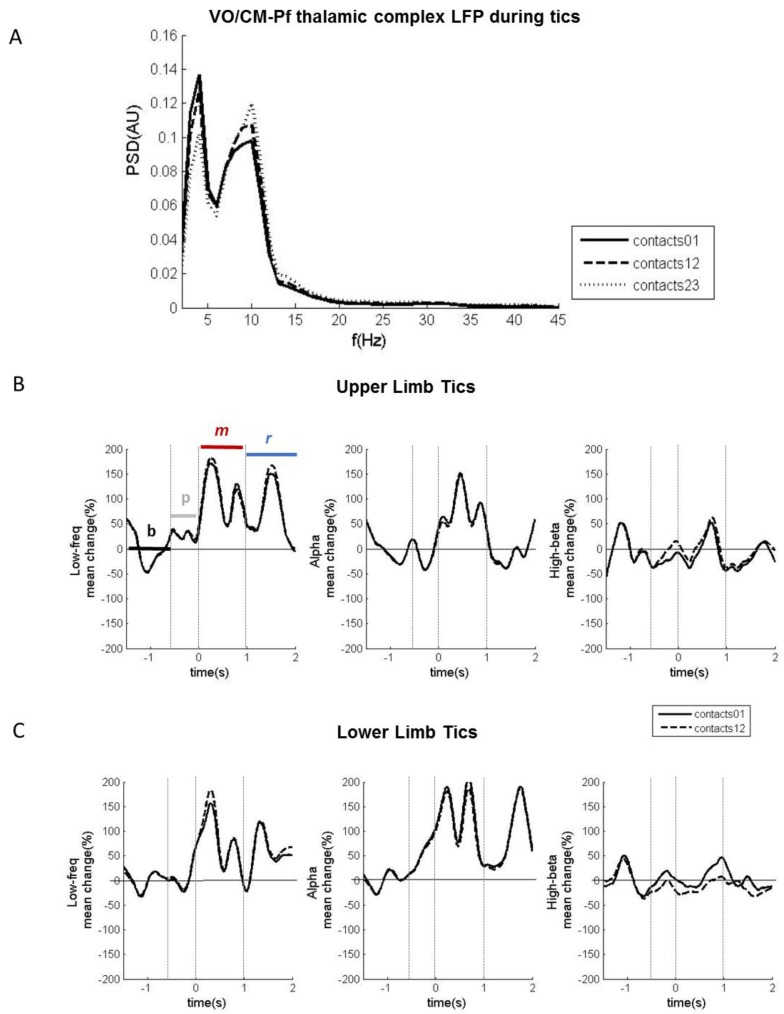
Vo/CM-Pf LFP oscillations during involuntary movements in one patient. (**A**) Power spectrum of LFPs captured from the Vo/CM-Pf nucleus using contacts 0–1 (black solid line), 12 (dashed black line), and 23 (dashed grey line) during motor tics. The *x*-axis represents frequency (Hz) and the *y*-axis represents the normalized power spectral density (PSD, arbitrary unit). (**B**) LFP power modulations for the low-frequency, alpha, and high beta bands recorded during upper limb tics from the electrode contact pairs 0–1 and 1–2, averaged across all observed tics. Power modulations are expressed as percentage changes from the baseline phase and were estimated starting 1.5 s before the movement onset until 2 s after the movement onset. The four movement-related phases were baseline (b), pre-movement (p), movement (m), and recovery (r). *X*-axis: time (s); *y*-axis: Hilbert-power modulations (%). (**C**) LFP power modulations for the low-frequency, alpha, and high beta bands recorded during lower limb tics from electrode contact pairs 0–1 and 1–2, averaged across all observed tics. The plots are organised as they are in panel B.

**Figure 4 brainsci-08-00004-f004:**
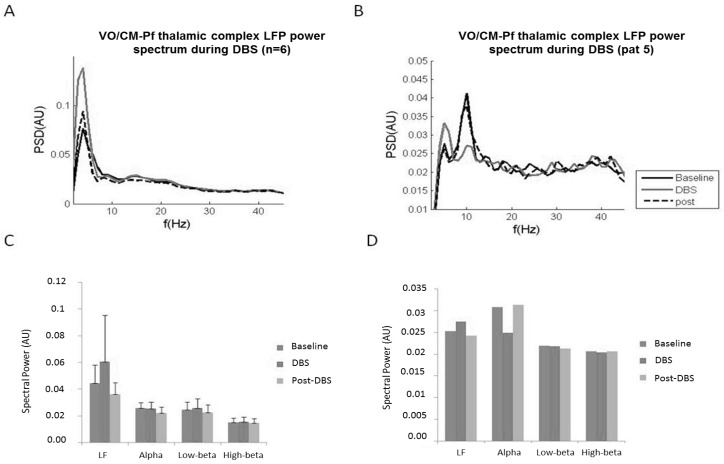
The LFP power spectrum obtained from the Vo/CM-Pf during ongoing DBS. (**A**) Mean power spectrum across all Vo/CM-Pf nuclei (*n* = 6) for LFPs at rest (black solid line), during ongoing DBS (grey solid line), and after DBS (black dashed line). The *x*-axis represents frequency (Hz) and the *y*-axis represents the normalized power spectral density (PSD, arbitrary unit). (**B**) LFP power spectrum from a single patient (case 5 in Table 3) during ongoing DBS. The plot is organized as in panel A. (**C**) The histograms represent the mean LFP spectral power (*n* = 6) for low-frequency, alpha, low beta, and high beta bands during the three experimental conditions: baseline (dark grey), DBS (middle grey), and post-DBS (light grey). Error bars represent standard errors. (**D**) The histograms represent the LFP spectral power in case 5. The histograms are organised as they are in panel C.

**Figure 5 brainsci-08-00004-f005:**
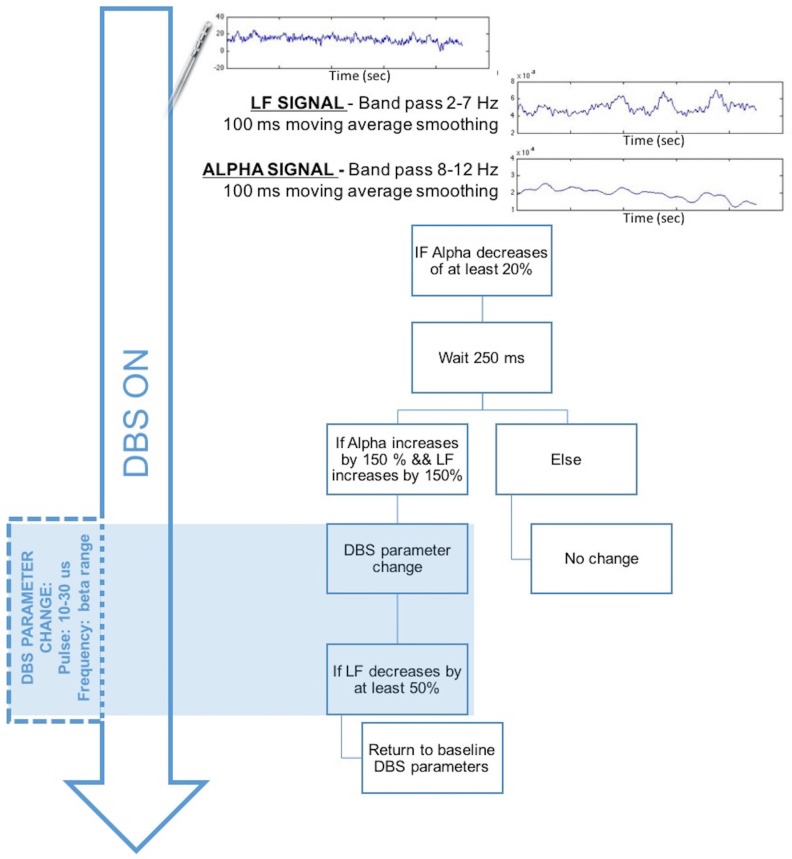
Hypothesized adaptive feed-forward strategy for fast changes in deep brain stimulation (DBS) parameters to control tics. LF = low-frequencies band (2–7 Hz), ALPHA = alpha band (8–13 Hz).

**Table 1 brainsci-08-00004-t001:** Demographic and deep brain stimulation (DBS) details for the patients recorded during the acute phase.

Patient ID	Gender	Age (Years)	Severe Psychiatric Comorbidity	DBS Target	Preoperative Assessment		12 Months Assessment	
					YBOCS	YGTSS	YBOCS	YGTSS
1	m	39		Vo/CM-Pf	12	75	10	40
2	m	42		Vo/CM-Pf	12	79	11	70
3	f	47		Vo/CM-Pf	17	45	20	29
4	m	40		Vo/CM-Pf	3	65	5	35
5	m	27		Vo/CM-Pf	17	28	0	14
6	m	24	x	NA	23	79	11	48
7	f	29	x	NA	37	70	30	70

m = male. f = female. DBS = deep brain stimulation. YBOCS = Yale-Brown Obsessive-Compulsive Scale; YGTSS = Yale Global Tic Severity Scale.Vo/CM-Pf = Ventralis Oralis Centro Median Parafascicular thalamic nucleus. NA = Nucleus Accumbens.

**Table 2 brainsci-08-00004-t002:** Summary of average LFP modulations observed in the Vo/CM-Pf thalamic nucleus during voluntary and involuntary movements in the low-frequency (2–7 Hz) and alpha (8–13 Hz) bands in patients with TS.

Movement Type		Low-Frequency (2–7 Hz)			Alpha (8–13 Hz)		
		Onset (ms)	Duration (ms)	% Change	Onset (ms)	Duration (ms)	% Change
**Voluntary movement**	Pre-movement [−500 0] ms	−250	250	+40%	−250	250	+20%
	Movement [0 1000] ms	0	800	+60%	0	500	+60%
	Recovery [1000 2000] ms	800	1200	+20%	500	1500	+20%
**Involuntary movement**	Pre-tic [−500 0] ms	-	-	-	−250	250	−20%
	Tic [0 1000] ms	0	1000	150%	0	1000	150%
	Recovery [1000 2000] ms	1000	2000	100%	-	-	-

**Table 3 brainsci-08-00004-t003:** Demographic and deep brain stimulation (DBS) details of the patients recorded in the chronic phase.

Patient	Gender	Age (Year)	Preoperative Assessment		12 Months Assessment		DBS Target	Chronic Stimulation for (Years)	Recording Contacts		Impedance of the Recording Contacts (kΩ)
			YBOCS	YGTSS	YBOCS	YGTSS					
1	m	31	32	92	18	40	Vo/CM-Pf	7	Right	46	4.5
									Left	02	5.1
2	m	35	21	89	14	38	Vo/CM-Pf	5	Right	57	3
									Left	02	6.2
3	m	48	20	78	14	30	Vo/CM-Pf	2	Left	12	7.6
4	f	25	25	78	19	40	Vo/CM-Pf	N/A	Right	46	4.7
									Left	02	3.7
5	m	38	30	42	26	26	Vo/CM-Pf	N/A	Right	46	7
6	m	24	0	28	2	25	Vo/CM-Pf	1	Right	47	0
									Left	03	4.9
7	f	39	5	56	11	31	Vo/CM-Pf	4	Right	46	5
									Left	02	7.3
									Right	46	3.8
8	m	29	38	69	21	20	NA	3	Left	02	2.2

m = male. f = female. DBS = deep brain stimulation. YBOCS = Yale-Brown Obsessive-Compulsive Scale; YGTSS = Yale Global Tic Severity Scale.Vo/CM-Pf = Ventralis Oralis Centro Median Parafascicular thalamic nucleus. NA = Nucleus Accumbens.N/A = Not Available.

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
