# Peer review of "Adaptive Deep Brain Stimulation (aDBS) for Tourette Syndrome"

_brainsci, 2017, doi:10.3390/brainsci8010004_

Round 1
Reviewer 1 Report
In this article entitled Adaptive deep brain stimulation (aDBS) for Tourette 3 syndrome?
the authors perform different experiments to record LPF’s from thalamus and NA in patients with Tourette’s syndrome. Although the hypotheses of the study are very relevant for the field of DBS and TS, there are some limitations that would need to be addressed.
Tics are marked by a premonitory urge that results in a semi-involuntary to relief the urge or discomfort that precedes. The authors classified tics as involuntary movements and did not mention any correlation between the LFP recordings with the premonitory symptoms.
The way the manuscript is written is out of standards. There is an introduction and then the experiments are mixed with review of literature and discussion. In some parts of the paper, the reader can become confused if a certain experiment described was performed by the authors or published in a previous literature. I would suggest that the structure of the papers is reformulated for clarity.
The authors state in the paper that DBS targets in TS are chosen based not only in tics but also in psychiatric comorbidity. The psychiatric aspects appear to be neglected when aDBS is proposed based on recordings of LFP associated with tics. The limitations of this approach with respect to management of psychiatric symptoms need to highlighted in the manuscript.
In the acute phase, only one patient presented with tics, which limits the interpretation of the results. In the chronic phase, it is unclear how many patients were recorded during the tics. The authors would need to clarify how many patients were recorder in each phases of the experiments.
At the end of the paper, the authors propose an algorithm for aDBS in TS, however they do not specify the targets. According to their findings, GPI versus thalamic DBS have very different physiological signatures during tics, therefore the algorithm proposed would probably not fit in both targets. Also, we know that there is large variability in lead location in different centers, and a description (table) of exact lead locations would strengthen the findings. Also, lead models used were not reported.
The conclusions of the study probably overestimated the findings from their experiments, and should be proposed with more caution, and emphasizing the need for further studies.
Finally, ethical aspects of the research, such as IRB approval and informed consent were not reported.
Minor issues:
Line 28 – LFP’s abbreviation appear before description of the term
Line 66 – High frequency stimulation is > 100 Hz – it is written <.
The text between lines 120 and 125 need to be rewritten, particularly the frequency bands could be better defined. For a reader that is not experienced in neurophysiology, these concepts can be confusing.
Lines 143 -147 also need review for improvement of readability.
I would encourage a review in the title, which does not reflect the experiments performed in the study.
Author Response
REPLY TO REVIEWERS
MS # brainsci-225267
REVIEWER 1
In this article entitled Adaptive deep brain stimulation (aDBS) for Tourette 3 syndrome?
the authors perform different experiments to record LPF’s from thalamus and NA in patients with Tourette’s syndrome. Although the hypotheses of the study are very relevant for the field of DBS and TS, there are some limitations that would need to be addressed.
Tics are marked by a premonitory urge that results in a semi-involuntary to relief the urge or discomfort that precedes. The authors classified tics as involuntary movements and did not mention any correlation between the LFP recordings with the premonitory symptoms.
Reply: We agree with the reviewer that this is an interesting point to be considered. Unfortunately, there are no LFP studies directly evaluating premonitory urges and our patient showing tics after DBS did not report this sensation during experiments.
We included a comment on this point in the “LFPs during Involuntary Movements” section, adding: “Also, the presence of sensory phenomena of feeling a “urge to move” (premonitory urge) that are relieved by tic expression has been documented as a distinct feature of TS (33). This would suggest the need to study the neurophysiological correlates of such phenomena to better understand TS tics phenomenology. However, the premonitory urge was mainly associated by neuroimaging studies to an activation of neocortical and paralimbic areas, with some activity also in the putamen, caudate, and claustrum (34) and there are no LFP studies specifically reporting observations on premonitory urge.” (lines 236-242)
And in the Discussion: “Also, tics phenomenology should take into account the presence of a premonitory urge (33) that would further enhance the ability of the feed-forward strategy to suppress tics. In fact, the premonitory urge, if recognized by neurophysiological recordings, may act as a trigger for specific DBS patterns aimed to suppress the urge that would be otherwise satisfied by tic expression, thus avoiding tic onset. This hypothesis is at present purely speculative and would require targeted experiments that specifically study the LFP signature (if any) of premonitory urge and the ability of DBS to satisfy the urge.” (lines 368-374)
The way the manuscript is written is out of standards. There is an introduction and then the experiments are mixed with review of literature and discussion. In some parts of the paper, the reader can become confused if a certain experiment described was performed by the authors or published in a previous literature. I would suggest that the structure of the papers is reformulated for clarity.
Reply: This is a review paper in which we are trying to discuss with the available data whether or not LFP-based aDBS in Tourette is feasible. To further support this discussion, we included some additional experiments on LFP recordings that complement the limited literature available so far.
We therefore clarified this approach in the Abstract by changing:
“To increase such knowledge, we recently acquired and analyzed, in patients with TS after DBS electrode implantation, LFPs at rest, during voluntary and involuntary movements (tics) and during ongoing DBS. We enrolled patients with severe TS, satisfying DSM-IV-TR, refractory to standard drug treatment: patients with motor tics were implanted in Vo/CM-Pf nucleus and patients with motor tics and severe psychiatric comorbidities in the NA. The recording sessions included: (1) LFP at rest; (2) LFP for the time required to observe at least 15 tics; (3) LFP during 15 voluntary movements; (4) LFP during ongoing DBS for 6 minutes. We calculated LFP power spectral density for different frequency bands: low-frequency (LF), alpha and beta bands. LFPs showed two main oscillatory activities, one in the Low Frequency (LF) band (2–7 Hz) and one in the alpha band (8–13 Hz). The most consistent power modulations during voluntary movements were in the LF and in the alpha bands: power increased started before movement onset. During tics LF and alpha power increased. Moreover, DBS induced specific LFP changes: LF power increased whereas the alpha band decreased from baseline during ongoing DBS and returned to baseline power after DBS was turned off.”
to
“In this paper, we review the available literature supporting the feasibility of the LFP-based aDBS approach in TS. In addition, to increase such knowledge, we report explorative findings on LFPs recently acquired and analyzed, in patients with TS after DBS electrode implantation, at rest, during voluntary and involuntary movements (tics) and during ongoing DBS.”; (lines 36-39)
and in the Introduction by changing
“Here we review the knowledge on LFP coding in TS, and add some new results, in order to propose a possible model for LFP-based aDBS for Tourette.”
to
“In this paper, we will provide a review on the available literature supporting the feasibility of the LFP-based aDBS approach in TS. To further support this hypothesis, we also report explorative findings on LFPs recently recorded in patients with TS after DBS electrode implantation. After a short explanation of the adaptive DBS concept guided by LFPs, the manuscript is organized in sections, each discussing the specific LFP signature in TS at rest, during voluntary and involuntary movements (tics), and during ongoing DBS. The final Discussion section summarizes the results coming from the literature and from the additional experiments here reported to answer the question of aDBS feasibility in TS.” (lines 89-96)
Finally, we highlighted the additional experiments in each paragraph including a subheading.
The authors state in the paper that DBS targets in TS are chosen based not only in tics but also in psychiatric comorbidity. The psychiatric aspects appear to be neglected when aDBS is proposed based on recordings of LFP associated with tics. The limitations of this approach with respect to management of psychiatric symptoms need to highlighted in the manuscript.
Reply: In the Discussion, we added “Finally, this first exploratory model for aDBS is based on data coming from the thalamus, thus not taking into account psychiatric comorbidities that are observed in a large number of TS patients (37) and that lead to the recent characterization of the Obsessive-Compulsive Tic Disorder (OCTD) specifically addressing OCD and TS comorbidity (38). The patients here analyzed were, for the most part, implanted in the CM/Pf-VO thalamic complex, which was However, as well as in Parkinson’s disease, LFP studies have first focused on motor symptoms and then on cognitive dysfunctions (22). Therefore, to propose an advanced aDBS model, new experiments specifically targeted to evaluating LFPs in domains as attention, memory, executive functions, language, motor and visuomotor functions, should be conducted, especially in patients implanted in targets as the anterior GPi (38).” (lines 375-383)
In the acute phase, only one patient presented with tics, which limits the interpretation of the results. In the chronic phase, it is unclear how many patients were recorded during the tics. The authors would need to clarify how many patients were recorder in each phases of the experiments.
Reply: Tables 1 and 3 report the number of patients recorded in the two conditions, 7 patients and 8 patients respectively. No patients were recorded during tics in the chronic phase. We clarified this point in the text adding “LFP recordings took place in the surgery room, while lead extensions were exposed before the connection to the new implanted stimulator. Therefore, the experimental setting was limited to rest recordings (no movement-related and no tics-related LFPs were recorded) with and without DBS turned ON.” (section: Chronic LFP recordings during DBS, lines 292-295)
At the end of the paper, the authors propose an algorithm for aDBS in TS, however they do not specify the targets. According to their findings, GPI versus thalamic DBS have very different physiological signatures during tics, therefore the algorithm proposed would probably not fit in both targets. Also, we know that there is large variability in lead location in different centers, and a description (table) of exact lead locations would strengthen the findings. Also, lead models used were not reported.
Reply: As suggested by the reviewer, we clarified, in the Discussion, that the algorithm proposed is based mainly on the data from the thalamus, and that aDBS for other targets should be designed after other experiments. We added “These results, however, were obtained mainly in the thalamus, that was one of the first targets explored for TS DBS, thus limiting the possibility to design LFP-based aDBS algorithms dedicated to other targets that will require further experiments.” (lines 325-327).
We also underlined the speculative nature of our algorithm adding: “If the observations available in our limited sample will be confirmed in larger studies, we may speculate that, in patients implanted with thalamic DBS, monitoring thalamic Alpha and LF power changes in windows of 250 ms should be sufficient to detect a significant change in the oscillatory pattern able to identify a tic and apply a DBS parameter change. As an example of the possible applicability of the LFP-based aDBS approach, Figure 5 shows an adaptive feed-forward strategy that changes DBS parameters when a possible tic onset is detected, according to the available LFP characteristics during involuntary movements. In this speculative hypothesis, ... ” (lines 341-347)
The surgical procedure is the same as described in Marceglia et al, Mov Disord 2010 and in Servello et al, JNNP 2008. We added these two references and the electrode model as requested (lines 158-160).
The conclusions of the study probably overestimated the findings from their experiments, and should be proposed with more caution, and emphasizing the need for further studies.
Reply: As suggested, we toned down the conclusions, underlining the limitations of the current approach that are now phrased as “In conclusion, the present review, enriched by some original results, provides some indications that supports the hypothesis that LFP-based aDBS in Tourette, at least in the thalamus, is feasible. Further studies in patients are needed to better understand the LFP signature of psychiatric comorbidities and non-tic disease manifestations, and to include other DBS target that are now considered very promising for treating TS, such as the anterior GPi.” (lines 385-389)
Finally, ethical aspects of the research, such as IRB approval and informed consent were not reported.
Reply: Done (lines 159-161 and lines 289-291)
Minor issues:
Line 28 – LFP’s abbreviation appear before description of the term
Reply: Done
Line 66 – High frequency stimulation is > 100 Hz – it is written <.
Reply: Done
The text between lines 120 and 125 need to be rewritten, particularly the frequency bands could be better defined. For a reader that is not experienced in neurophysiology, these concepts can be confusing.
Reply: Done (lines 116-123)
Lines 143 -147 also need review for improvement of readability.
Reply: Done (lines 141-146)
I would encourage a review in the title, which does not reflect the experiments performed in the study.
Reply: Since the aim of the paper is to review the available literature to explore the possibility of LFP-based aDBS in Tourette, we would prefer to leave the main question in the title.
Reviewer 2 Report
Adaptive deep brain stimulation (aDBS) for Tourette Marceglia et al This is an interesting paper. The authors have performed LFP recordings in patients having DBS for Tourette Syndrome. Recordings were made mainly from the Vo/CM-Pf nucleus in the thalamus, also in 3 patients from the N Accumbens. They recorded at rest, during voluntary movement and during tics. In a separate set of patients they recorded during DBS. Using spectral analysis of Local Field Potentials they show activity mainly in the low frequency and alpha bands. During voluntary movement power in both bands increased during pre-movement, movement and recovery. During tics low frequency increased in movement and recovery. Alpha reduced in pre-movement Alpha increased during movement and recovery for lower limbs, only in movement for upper limbs. During DBS 6 patients had increase low frequency. 1 patient also reduced alpha. The authors discuss the possibility that these findings could help with adaptive DBS. This is an interesting paper which I feel is worthy of publication. However I have several comments / concerns. The participants numbers are fairly low, in some cases very low (eg one participants only recorded during tics). No statistical analysis has been performed. The results are therefore purely observational and I feel the authors should therefore be much more cautious in interpreting them. For example looking at Figure 4, the error bars are very large and I doubt these changes would be statistically significant. The authors mention a decrease in alpha in the premotor phase before a tic. This is perhaps the most interesting finding of the paper, but for some reason is not mentioned in the text of the results, only in the table then discussion. Although at the same time, looking at Figure 3, the decrease in alpha was only seen with upper limb tics, not with lower limb tics. Also only in the one patient that was studied. Therefore this finding must be considered very exploratory. I therefore don’t think the authors are justified in basing their whole adaptive DBS strategy on this putative alpha reduction. Figure 2B, third plot is supposed to be high beta but the plot says alpha. In summary, I feel this is an interesting and worthwhile study. However the results must be considered very exploratory and I think the hypothesised feed forward adaptive DBS strategy is too big a leap to include at this stage.Author Response
REPLY TO REVIEWERS
MS # brainsci-225267
REVIEWER 2
Adaptive deep brain stimulation (aDBS) for Tourette Marceglia et al This is an interesting paper. The authors have performed LFP recordings in patients having DBS for Tourette Syndrome. Recordings were made mainly from the Vo/CM-Pf nucleus in the thalamus, also in 3 patients from the N Accumbens. They recorded at rest, during voluntary movement and during tics. In a separate set of patients they recorded during DBS. Using spectral analysis of Local Field Potentials they show activity mainly in the low frequency and alpha bands. During voluntary movement power in both bands increased during pre-movement, movement and recovery. During tics low frequency increased in movement and recovery. Alpha reduced in pre-movement Alpha increased during movement and recovery for lower limbs, only in movement for upper limbs. During DBS 6 patients had increase low frequency. 1 patient also reduced alpha. The authors discuss the possibility that these findings could help with adaptive DBS. This is an interesting paper which I feel is worthy of publication. However I have several comments / concerns.
Reply: We thank the reviewer for the positive consideration of our work.
The participants numbers are fairly low, in some cases very low (eg one participants only recorded during tics). No statistical analysis has been performed. The results are therefore purely observational and I feel the authors should therefore be much more cautious in interpreting them. For example looking at Figure 4, the error bars are very large and I doubt these changes would be statistically significant.
Reply: We added in the discussion a section on study limitations: “However, this is only a theoretical speculation based on the literature and on the current results on tic detection and DBS effects over LFPs recorded in the thalamus in Tourette patients and has several important limitations. First, the neurophysiological observations underlying the adaptive model are on a limited number of patients, and should be confirmed by larger studies before being considered as conclusive. Also, tics phenomenology should take into account the presence of a premonitory urge (33) that would further enhance the ability of the feed-forward strategy to suppress tics. In fact, the premonitory urge, if recognized by neurophysiological recordings, may act as a trigger for specific DBS patterns aimed to suppress the urge that would be otherwise satisfied by tic expression, thus avoiding tic onset. This hypothesis is at present purely speculative and would require targeted experiments that specifically study the LFP signature (if any) of premonitory urge and the ability of DBS to satisfy the urge.
Finally, this first exploratory model for aDBS is based on data coming from the thalamus, thus not taking into account psychiatric comorbidities that are observed in a large number of TS patients (37) and that lead to the recent characterization of the Obsessive-Compulsive Tic Disorder (OCTD) specifically addressing OCD and TS comorbidity (38). The patients here analyzed were, for the most part, implanted in the CM/Pf-VO thalamic complex, which was However, as well as in Parkinson’s disease, LFP studies have first focused on motor symptoms and then on cognitive dysfunctions (22). Therefore, to propose an advanced aDBS model, new experiments specifically targeted to evaluating LFPs in domains as attention, memory, executive functions, language, motor and visuomotor functions, should be conducted, especially in patients implanted in targets as the anterior GPi (38).” (lines 364-383)
The authors mention a decrease in alpha in the premotor phase before a tic. This is perhaps the most interesting finding of the paper, but for some reason is not mentioned in the text of the results, only in the table then discussion.
Reply: We added “The alpha band also showed a desynchronization in the pre-movement phase, particularly in the upper-limb tics slightly observable also in the lower-limb tics, even though anticipated” (lines 262-264)
Although at the same time, looking at Figure 3, the decrease in alpha was only seen with upper limb tics, not with lower limb tics. Also only in the one patient that was studied. Therefore this finding must be considered very exploratory. I therefore don’t think the authors are justified in basing their whole adaptive DBS strategy on this putative alpha reduction.
Reply: We agree with the reviewer that this is a speculative assumption. We mentioned it in the Paragraph dedicated to the limitations of the observations and we also toned down the aDBS hypothesis changing the sentence “Taken together, all these considerations suggest that monitoring thalamic Alpha and LF power changes in windows of 250 ms should be sufficient to detect a significant change in the oscillatory pattern able to identify a tic and apply a DBS parameter change, as shown in Figure 5.” To “If the observations available in our limited sample will be confirmed in larger studies, we may speculate that, in patients implanted with thalamic DBS, monitoring thalamic Alpha and LF power changes in windows of 250 ms should be sufficient to detect a significant change in the oscillatory pattern able to identify a tic and apply a DBS parameter change. As an example of the possible applicability of the LFP-based aDBS approach, Figure 5 shows an adaptive feed-forward strategy that changes DBS parameters when a possible tic onset is detected, according to the available LFP characteristics during involuntary movements. In this speculative hypothesis, DBS may be delivered continuously…” (lines 341-348)
Figure 2B, third plot is supposed to be high beta but the plot says alpha.
Reply: We thank the reviewer for the observation. We realized that we have repeated the Alpha plot twice. We provided the correct plot in the revised Figure 2.
In summary, I feel this is an interesting and worthwhile study. However the results must be considered very exploratory and I think the hypothesised feed forward adaptive DBS strategy is too big a leap to
include at this stage.
Reply: We agree with the reviewer on the purely speculative nature of the hypothesized strategy. We would however prefer to leave it as a practical example of the feasibility of the aDBS approach. We have therefore rephrased the description of Figure 5, highlighting that it is only an exemplary application of how LFP-based aDBS can be implemented according to neurophysiological observations.